# Effective Optical Scattering Range Determination Based on Ray Tracing

Rongkuan Leng [1,2] , Zhiwei Chen [1,2], Shang Wang [1], Zhi Wang [1,3,*] and Chao Fang [1,*]

1. Changchun Institute of Optics, Fine Mechanics and Physics, Chinese Academy of Sciences, Changchun 130033, China
2. University of Chinese Academy of Sciences, Beijing 100049, China
3. School of Fundamental Physics and Mathematical Science, Hangzhou Institute for Advanced Study, UCAS, Hangzhou 310024, China
* Correspondence: wz070611@126.com (Z.W.); fangchao@ciomp.ac.cn (C.F.)

**Abstract:** Surface imperfections or contamination on an optical smooth surface is usually inevitable and causes scattering. The directional information of the scattered ray can be related to the spatial frequency through the grating equation. On the other hand, the layout of the optical system determines whether a scattered ray will finally reach the detector according to the positions and directions of the scattered rays. Therefore, the scattering propagation in an optical system is usually band-limited and the effective optical range differs for different systems. In this paper, a method based on ray tracing is described that can statistically determine the optical scattering and the band of roughness. The results can be an essential reference for optical surface polishing and contamination control.

**Keywords:** optical scattering; surface roughness; stray light analysis

## 1. Introduction

Micro-irregularities are distributed over the entire optical surface and scatter the incident light, which is inevitable even for high-quality optics. The scattering caused by surface profile height is well discussed in [1–4]. The statistical randomness of an optical surface can be expressed by the power spectral density function (PSD). It gives the information on the surface error as a function of the spatial frequency and relies highly on the manufacturing of the optical surface. For example, for a periodic irregularity produced by single-point diamond turning, the PSD gives the information about the tool feed rate, depth of cut, and vibration of the machine [5]. Furthermore, the scattering of an optical smooth surface can be predicted with the Rayleigh–Rice vector perturbation theory which relates the scattered power to the surface power spectral density [6].

However, in real scenarios, the optical system selects the scattered rays according to their positions and directions. The position information can allow the estimation of the scattering flux in combination with the incident radiance and scattering response of the surface [7]. The direction information can be used to determine the spatial frequency with the hemispherical grating equations. The scattering that is able to reach the detector becomes stray light and limits the resolution of the image [8,9]. The pattern is usually caused by the scattering near the specular direction which is called small-angle scattering. In comparison, the scattering that cannot reach the detector of the system results in energy loss and reduces the contrast of the image [10]. This process is related to the wide-angle scattering which corresponds to the high spatial frequency region of PSD [10]. The band of delivered scattering differs for different optical elements even though the optical surfaces have the same topography. Therefore, due to the stochastic nature of the scattering and the diversity of the optical systems, the numerically intensive ray tracing method is a perfect method to determine the cutoff frequency. However, the combination of the surface features, optical scattering and ray tracing theory is not yet well discussed. In this paper,

the classic optical scattering is combined with the ray tracing theory to determine both the system reaction of scattering and band of surface imperfections. Relevant definitions and the thermotical background are introduced in Section 2. In Section 3, two typical surface features are traced in order to determine the effective range of the surface imperfections. The methods can determine the critical spatial frequency range of roughness which can instruct the manufacturing processes.

Moreover, local defects such as contamination are also considered. Traditionally, the local defects are characterized by the cleanness level discussed in [11] where the considered particles are idealized and scattering loss is statistically related to the cleanness. However, the effect of a single defect is usually hard to measure or quantify. Based on the ray tracing method discussed in this paper, the influence of a single imperfection is also analyzed which can inversely instruct the configuration of scattering measurement. The method reveals the selective nature of an optical system from the statistical view and its great flexibility.

## 2. Theoretical Background

The surface roughness that produces scattering which limits the performance of the imaging is our primary concern. The scattering depends on the wavelength, polarization and the distribution of the incident beam. The surface topography also influences the scattering distribution and in Section 2.1 the definition of roughness is introduced and Section 2.2, the relevant concept of classic optical scattering is given. Finally, the scattering transfer behavior in a typical system is demonstrated which is the center of this paper.

### 2.1. Roughness

Roughness is a common measure of surface topographic irregularities. Most of the topography measurements are in the form of digitized data of the surface height. Therefore, there is no unique roughness value for a surface. It depends on the length of the surface profile L, lateral resolution and the sampling distance d. The root-mean-square roughness $\sigma_{rms}$ is most used surface statistical parameter and is defined as:

$$\sigma_{rms} = \sqrt{\frac{1}{N} \cdot \sum_{i=1}^{N} (z_i(x) - \bar{z}(x))^2} \tag{1}$$

where N is the number of samplings. $z_i(x)$ is the surface height of a single sampled point. $\bar{z}(x)$ is the mean surface height. The same surface can have many different values depending on the measurements. The autocovariance function is a measure of the correlation and periodicity of the surface roughness. It is defined as the average of the deviation of mean surface height multiplied by a translated version of itself:

$$G(\tau) = \lim_{L \to \infty} \frac{1}{L} \int_{-L/2}^{L/2} [z(x) - \bar{z}(x)][z(x+\tau) - \bar{z}] \, dx \tag{2}$$

where $\tau$ is the lag length. For zero lag length, the autocovariance function becomes the squared roughness which is:

$$G(0) = \sigma^2 = \lim_{L \to \infty} \frac{1}{L} \int_{0}^{L} [z(x) - \bar{z}(x)]^2 \, dx \tag{3}$$

The power spectral density function is the frequency spectrum of the surface roughness. It is the square of the Fourier transform of surface profile, or the Fourier transform of surface autocovariance function:

$$PSD(f_x, f_y) = \lim_{L \to \infty} \frac{1}{L} \left| \int_{-L/2}^{L/2} z(x) \times \exp[-2\pi j \, f_x x] dx \right|^2 \tag{4}$$

where $f_k$ is the spatial frequency that fulfills:

$$\frac{1}{Nd} \le f_k = \frac{k}{Nd} \le \frac{1}{2d}, \ (1 \le k \le N/2) \tag{5}$$

where $1/Nd$ is usually called the fundamental frequency and $1/2d$ is the Nyquist or the foldover frequency, the highest allowed frequency. The sampling distance d should be as small as possible. If it is much larger than the lateral resolution, aliasing occurs and leads to failure of measurement.

*2.2. Scattering*

The scattering geometry is defined in Figure 1. The incident beam impinges the surface at the position of (x, y) with the angle of incidence $\theta_i$ and an illuminated area dA. The beam is scattered into a solid angle d$\omega$ with the polar angle $\theta_s$ and the azimuth $\varphi_s$. The scattered radiance is related to the incident irradiance with:

$$L(x, y, \theta_s, \varphi_s) = \frac{d^2\Phi_s(x, y, \theta_s, \varphi_s)}{d^2\Phi_{iN}(x, y, \theta_{in}) \cdot \cos(\theta_s) \cdot d\omega} \cdot \frac{d^2\Phi_{in}(x, y, \theta_{in})}{dA} \tag{6}$$

where $\Phi_s$, $\Phi_{in}$ is the scattered flux and the incident flux. The first term on the right-hand side is usually called the Bidirectional Scattering Distribution Function (BSDF) which is usually split into a reflected component (BRDF) and a transmitted component (BTDF). For an isotropic surface, BRDF exhibits rotational symmetry and can be reduced to an in-plane scattering [12].

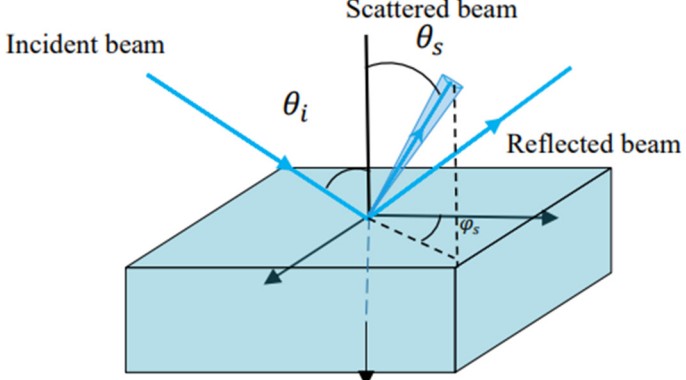

**Figure 1.** Scattering geometry for the definition of BRDF.

The radiometric definition of the scattering is straightforward. However, the accurate measurement and modeling can be difficult. The scattering signature is a result of roughness, surface structure, contaminations, scratches, wavelength, and polarization. Furthermore, the digital noise and optical alignment also make the signature even more complicated. Moreover, the scattering can be related to the surface PSD function with the Rayleigh–Rice vector perturbation theory:

$$BRDF(\theta_i, \theta_s, \varphi_s, \lambda) = \frac{16\pi^2}{\lambda^4} \cdot \cos(\theta_i) \cdot \cos(\theta_s) \cdot Q \cdot PSD(f) \tag{7}$$

where the Q is called the optical factor which is a function of the incident angle, the scattered angle, the complex refractive index of the surface and the state of polarization. For an isotropic, optically clean and smooth surface, the hemispherical grating equation can be used to evaluate the spatial frequency:

$$f = \frac{\sin\theta_s - \sin\theta_i}{\lambda} \tag{8}$$

The total integrated scattering (TIS) is defined as the ratio of light scattered into a hemisphere to the total light reflected by the surface. According to scalar scattering theory, the ratio of specular reflectance $R_s$ to the total reflectance R can be estimated from rms roughness:

$$\frac{R_s}{R} = \exp\left[\left(-\frac{4\pi}{\lambda}\sigma_{rms}\cos\theta_i\right)^2\right] \tag{9}$$

When it is normal incidence, Equation (9) can be rewritten in combination with TIS:

$$TIS = \frac{R - R_s}{R} \approx \left(\frac{4\pi\sigma_{rms}}{\lambda}\right)^2 \tag{10}$$

However, for a superpolished surface, $R_s$ /R is close to unity and difficult to measure. Therefore, it is reasonable to collect the scattered light instead of the loss of reflectance. After integrating hemispherical scattering, the TIS becomes:

$$TIS = \int\limits_0^{2\pi}\int\limits_0^{\pi/2} BRDF \cdot \sin(\theta_s)\cos(\theta_s)d\theta_s d\varphi_s \tag{11}$$

Figure 2 shows the relation between the fundamental concepts of roughness and optical scattering. The power spectral density function can be derived either directly from the surface profile through the square of the Fourier transform or indirectly from the optical scattering if the angular distribution of scattered light is measured. However, the surface profile cannot be obtained from the PSD because the phase information about the surface is lost. The rms roughness is the square root of the zeroth term in the autocovariance function. Alternatively, the rms roughness can be obtained by integrating the measured values of the angle-resolved scattering for all angles to yield the TIS and then using Equations (9) and (10) for estimation. The roughness derived through the TIS is band limited which is in the range of optical response.

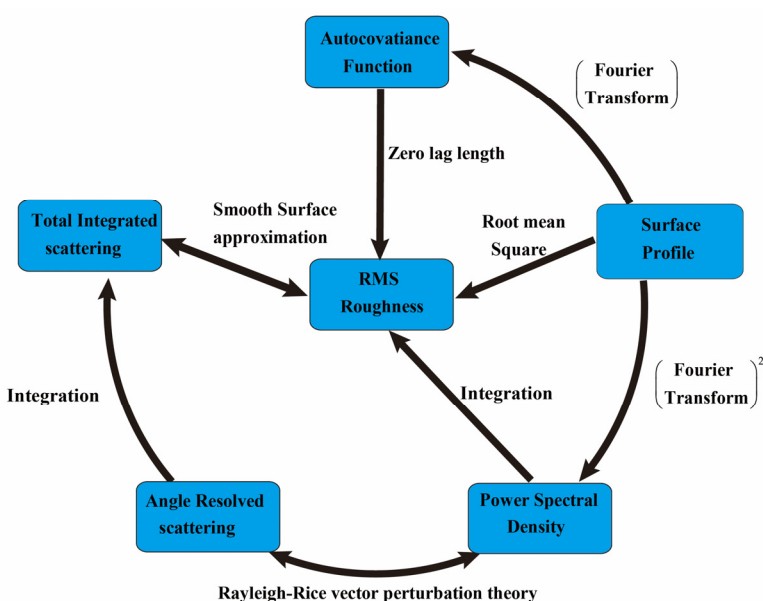

**Figure 2.** Relation between roughness and scattering parameters.

### 2.3. Ray Transfer in Optical Systems

The scattering reaches the detector and becomes stray light through three processes. First, the considered surface is illuminated and such a surface are often called an illuminated surface. Then, the incident light is scattered by the imperfections of the surface. Finally, the scattered rays are selected by the layout of the optical system according to its position and

directions. In other words, the illuminated surfaces need to be viewed by the detector [8]. For an infrared system, an additional source exists which is the internal emission of the sensor itself, which is often called thermal self-emission. Detailed analysis of such radiation is not the goal here. This paper focuses on relating the classical external stray light to the surface roughness, which is a concern for most optical systems. Such paths can be verified experimentally and precisely identified with the time of flight method which allows the determination of the origin of stray light by temporally discriminating the light at the detector during stray light testing [13]. The point spread function under the effect of surface error can be calculated as [14]:

$$\mathrm{PSF}_{\text{surface error}} = \mathrm{PSF}_0 * [\mathrm{SR} \cdot \delta(\alpha, \beta) + (\frac{\mathrm{TIS}}{\sigma_{\text{rms}}^2}) \cdot \mathrm{PSD}] \tag{12}$$

where $\mathrm{PSF}_0$ is the ideal point spread function, SR is the Strehl ratio, $\alpha, \beta$ are the direction cosines. In this way, the influence of surface error can be related to the image quality. In the optical design phase, only a few rays are considered to estimate the effect of aberration. However, due the stochastic nature of roughness, ray tracing is a more flexible method which directly reflects the acceptance–rejection nature of an optical system. Since the scattered rays are missed during the propagation, the ray transfer efficiency $\eta_r$ can be defined as:

$$\eta_r = \frac{\sum\limits_{i=1}^{N} V_i}{N_0} \tag{13}$$

In Equation (13), $N_0$ is the total number of considered rays. $V_i$ is the visibility term which equals 1 if the scattered ray reaches the detector or otherwise is 0. Because each scattered ray is uncorrelated, the process can be taken as a Bernoulli experiment. When $N_0$ approaches infinity, the ray transfer efficiency can be interpreted as the possibility that a scattered ray reaches the detector, which is:

$$\lim_{N_0 \to \infty} \mathbb{P}(\left| \frac{\sum\limits_{i=1}^{N} V_i}{N_0} - p_{0N} \right|) = 0 \tag{14}$$

$p_{0N}$ is the possibility of the scattered beam reaching the detector. Equations (13) and (14) reveal the selection of the scattering by the optical system. Moreover, with ray tracing, a preferred angular distribution of the scattering can be derived. In this way, the frequency range of each optical element can be derived. The cutoff frequency is an essential reference for figuring and polishing. For example, if a spectrum peak is not in the range of the optical response range of the system, the periodic structure most likely can be controlled. On the other hand, the ray distribution can be used to analyze the effect of a local defect. For example, the optical scattering of a scratch relies highly on the angle of incidence and the scattering angle. Measuring all of the situation can be difficult and unnecessary. Therefore, it is reasonable to determine the potential geometry and then measure only a few configurations, which will be discussed in the following section.

## 3. Application

In this section, a superpolished sample is considered. A series of measurements of surface features with different techniques is conducted in order to get precise topographic or statistical surface information. Then, the BRDF of the surface is introduced to a typical Cassegrain telescope. The effective directional range is recorded and the corresponding applications are shown in Section 3.2.

### 3.1. Sample

The sample used in this paper is made by Zerodur with magnetorheological finishing. A reflective silver-enhanced coating is used to achieve the high reflective performance in the wavelength of 1064 nm. The photograph of the superpolished sample is shown in Figure 3.

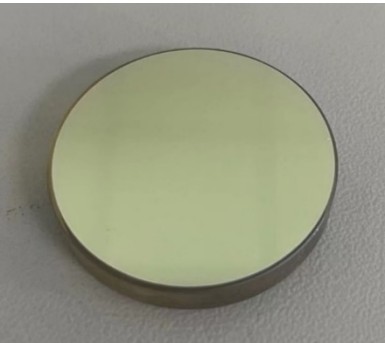

**Figure 3.** The superpolished sample with reflective silver-enhanced coating.

As mentioned before, there is no unique rms roughness for a surface. The measured results can be different for different instruments, sampling or frequency ranges. Therefore, a rigorous roughness characterization requires a combination of a variety of techniques. In order to get the topographic information, the sample is measured with an atomic force microscope (AFM, a field of 0.02 mm × 0.02 mm, 256 × 256 pixels). It works by scanning with a probe very close to the sample surface and measuring attractive or repulsive forces. The measurement result is shown in Figure 4a where the rms roughness equals 0.3 nm.

White-light Interferometry (WLI) is another typical method in microscopic profilometry. It is also used to measure the surface (10× and 50× objective). Figure 4b shows the profile measurement of the sample with 10× objective with WLI. Due to the short coherence length of the white light, only the area whose optical path is close to the interferometer arm length can be detected. The rms roughness in this case is 0.5 nm.

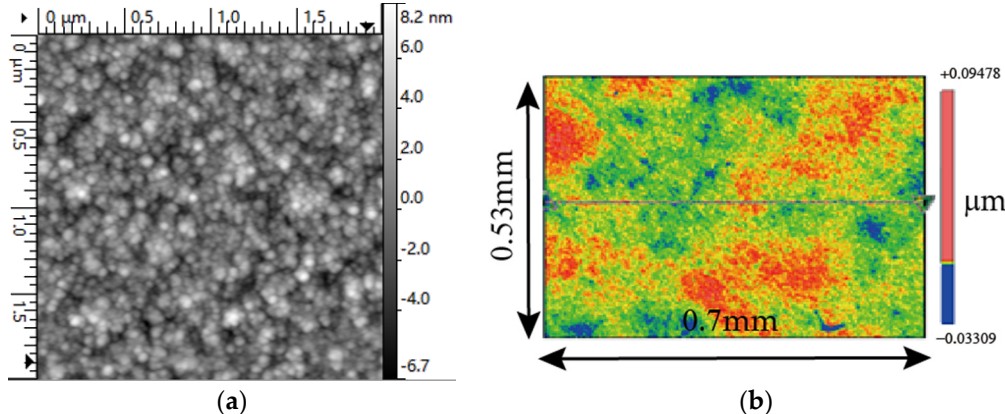

(**a**)  (**b**)

**Figure 4.** Topography measurement result of the superpolished sample shown in Figure 3. (**a**) The AFM measurement result, (**b**) the WLI measurement result with 10× objective.

The angle-resolved scattering measurement is a typical non-contact method. It has proven to be an effective method to determine surface roughness from the above discussion. An important condition is that, for roughness determination, the scattering should be solely caused by roughness. It should not be due to particulate contamination or local defects such as scratches, sleeks or digs. The sample is measured with a table-top system for light scatter measurement (Albatross-TT) [15]. The angles of incidence are 5°, 45° and 75°, respectively.

The scattered angle ranges from $-85°$ to $85°$ and the wavelength is 1064 nm. The results are presented in Figure 5.

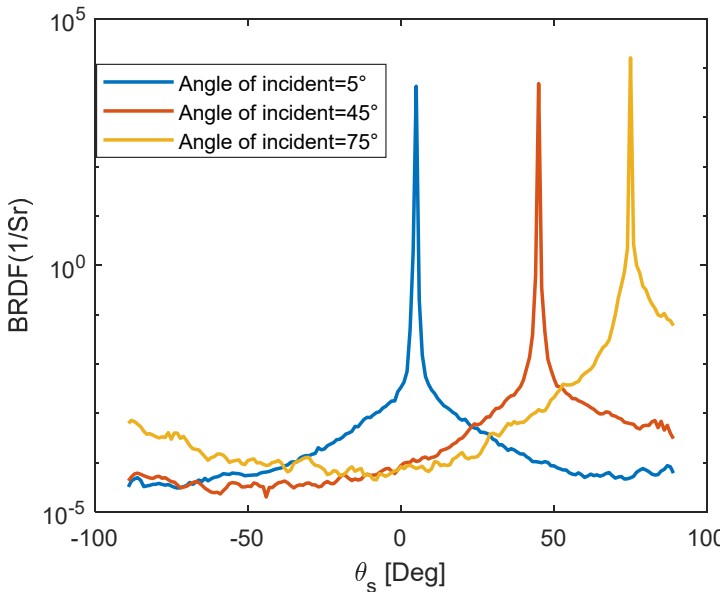

**Figure 5.** The BRDF measurement results under different angles of incidence.

The scattered measurement can be converted into the PSD data with Equation (7). Figure 6 is plotted considering all of the above-mentioned techniques. It can be seen that the PSDs under different techniques are in different frequency ranges. The relevant frequency ranges are shown in Table 1. The PSDs are in good agreement which will later be used for ray tracing.

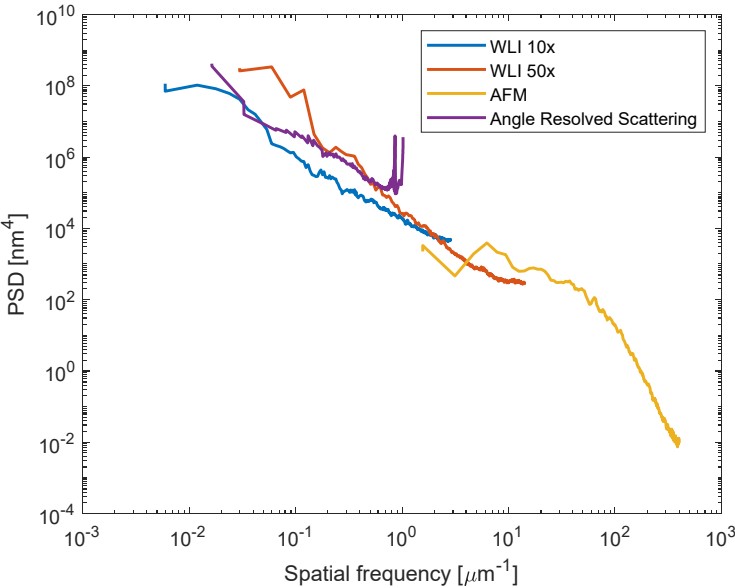

**Figure 6.** PSD measurement results with different techniques including WLI.

**Table 1.** Frequency range for different techniques.

| Technique | Frequency Range ($\mu m^{-1}$) |
| --- | --- |
| Angle-resolved scattering | 0.016–1.022 |
| WLI 10× | 0.0206–2.85 |
| WLI 50× | 0.0298–14.17 |
| AFM | 1.5–398 |

*3.2. Ray Tracing Result*

Optical scattering is a critical origin and is usually hard to remove when it is in the path of a specular beam. In this section, the stray light from scattering occurring in a typical Cassegrain telescope is considered. The optical configuration is listed in Table 2. The specular optical path is shown in Figure 7. The roughness and the scattering properties are given in Section 3.1. The simulation is achieved with numerically intensive software ASAP [16]. In the beginning, the roughness scattering is considered. With the directional information of the scattering, the effective frequency range can be derived. On the other hand, the scattering of local defects is also demonstrated. Such scattering is usually more critical and depends highly on the optical configuration such as incident angle, scattered angle and wavelength.

**Table 2.** Optical configuration of a typical Cassegrain system.

| Optical Element | Radius of Curvature (mm) | Semi-Diameter (mm) | Spacing (mm) |
|---|---|---|---|
| Primary | −40 | 5 | −15 |
| Secondary | −12.5 | 1.5 | 25 |
| Detector | ∞ | 0.5 | - |

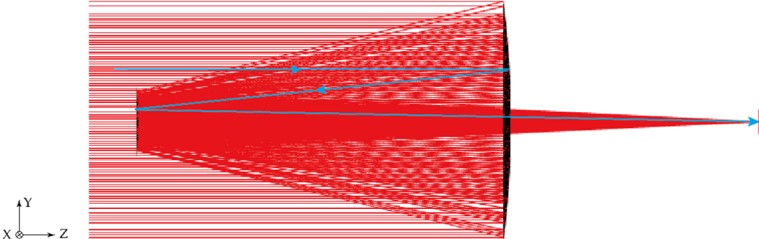

**Figure 7.** The optical path of the specular beam of a typical Cassegrain telescope.

When considering the scattering caused by roughness, the BRDF was introduced in the ray-tracing model and first-order scattering was computed for each mirror separately. The scattering of the primary and the secondary are separately considered. Figure 8a shows the scattering from the primary and Figure 8b shows the scattering path from the secondary. As can be seen, the scattering reaches the detector only when a certain distribution of direction and position is fulfilled. On the other hand, the roughness scattering is usually isotropic due to the random nature of the surface texture and the axis-symmetry of the Cassegrain telescope.

Because the local surface normal of a curved surface is different from different positions, it is reasonable to consider the angular deviation of the scattering path from the specular path. It makes sense because the BRDF of an isotropic surface is shift-invariant. Then, the BRDF curve should be shifted so that that the peak or the zero-deviation position is always in the direction of the specular path. A total of 30,000 scattered rays are picked out which originate from the primary and the secondary and reach the detector. For unit incident power, the scattering flux from the primary that reaches the detector is $1.05 \times 10^{-5}$ W and for the secondary is $8.5 \times 10^{-5}$ W. The deviation of each ray is recorded and plotted in Figure 9. which is the probability distribution indicating the angular distribution of the scattering that hits the detector. The green curve represents the deviation of the scattering from the primary while the orange represents the scattering from the secondary. It can be seen from Figure 9 that the scattering is evenly distributed around the specular beam for a rotationally symmetrical system. Furthermore, the histogram clearly shows that the scattering from the primary is more concentrated. The standard deviation of the scattering is $0.13°$ for the primary while for the secondary, the standard deviation is $0.56°$. According to Equations (13) and (14), the ray transfer efficiency for the scattering of the primary is $2 \times 10^{-5}$ while that for the secondary is $4 \times 10^{-4}$. It shows that the scattering from the

secondary has a larger possibility to reach the detector. The scattering from the primary is selected and only the scattering that is close to the specular light can reach the detector.

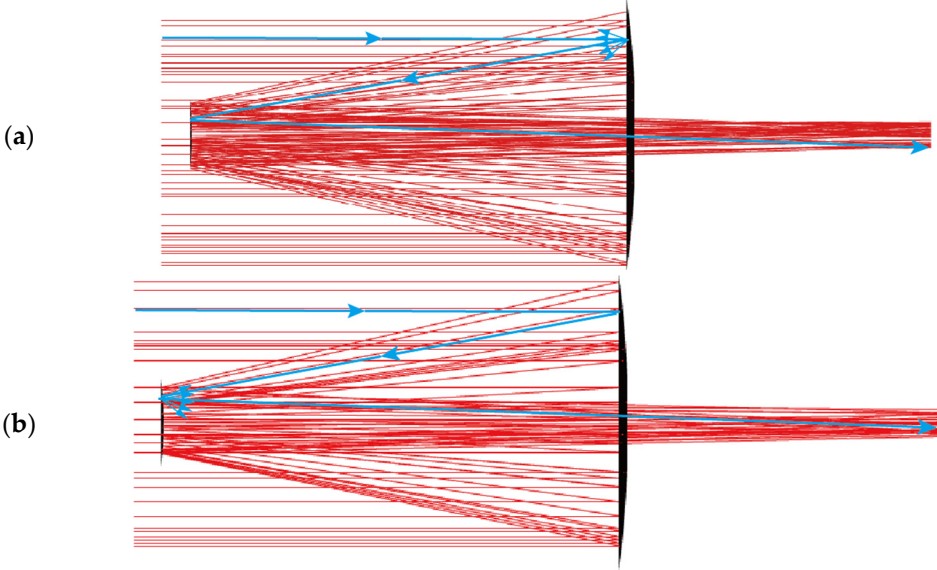

**Figure 8.** The paths of scattering that are from the primary (**a**) and the secondary (**b**) of the Cassegrain telescope.

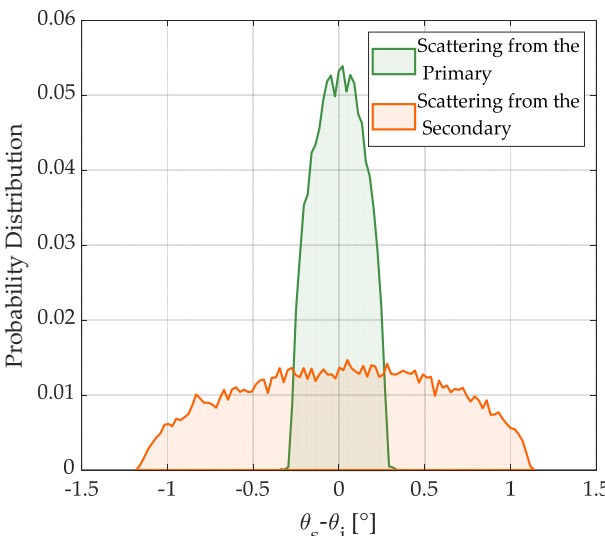

**Figure 9.** The probability distribution of the scattered ray deviation from the specular reflected direction.

On the other hand, according to Equation (8), the scattered angle and the incident angle are converted into the spatial frequency which is shown in Figure 10. The dots are for the measurement data from Section 3.1 and the blue curve represents the fitted curve. The maximum spatial frequency for the primary is 0.007 $\mu m^{-1}$ which is shown in the orange area in Figure 10. The green area is the spatial frequency range for the secondary with the maximum 0.02 $\mu m^{-1}$. Both Figures 9 and 10 indicate the different characteristics in stray light contribution for the primary and the secondary even through the roughness and the scattering properties are assumed.

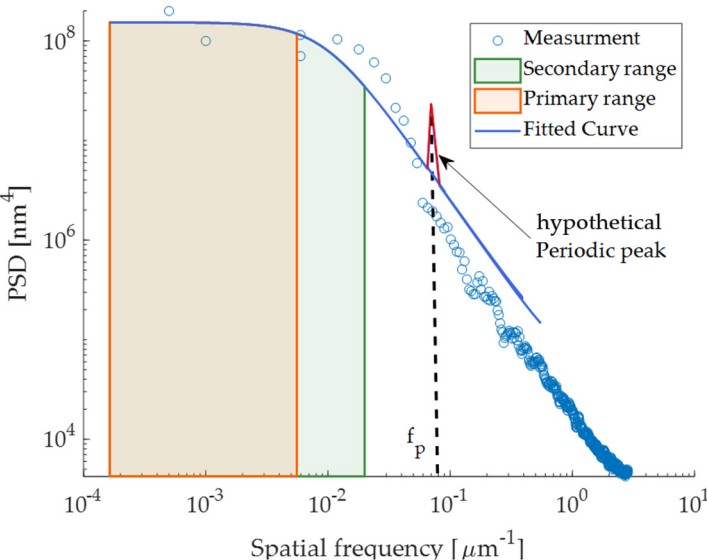

**Figure 10.** The effective frequency range of the PSD for the primary and the secondary.

Determining the spatial frequency range through ray tracing can also be related to the fabrication. For example, with a hypothetical periodic structure with the spacing $1/f_p$ on the surface, which is usually due to processing, a peak can be witnessed from the PSD curve which is shown as the red curve in Figure 10. Even though the periodic imperfection can cause small-angle scattering, it will not contribute to the in-field scattering because fp is neither in the frequency range of the primary nor that of the secondary. In other words, the scattering caused by such a kind of structure can be blocked or absorbed with proper mechanical design or black surface treatment.

A single local defect such as small particles scatters the incident beam differently and such a surface usually suffers from a higher scattering loss. For such a case, the clean surface assumption is not fulfilled and the Rayleigh–Rice vector perturbation theory is no longer valid. Figure 11a shows the scattering image of a particle on the considered surface. In comparison, Figure 11b is the instrument signature where the roughness is the only source of scattering. The measurement is conducted with a highly sensitive optical roughness sensor (Horos) [17,18] which can record the scattering image and the BRDF value simultaneously. The central peak is the specular reflected beam and the surroundings are the scattered light. The diffraction caused by the particle is also considered. Typically, the scattering caused by particles is usually anisotropic and a thorough measurement can be difficult and time-consuming. On the other hand, the scattering caused by the same particle might be different for optically different elements and positions. Analytical calculation can be difficult, making ray tracing an extremely flexible and practical method.

In order to analyze the effect of the local particles, three particles are introduced on the primary with the X-Y plane coordinates of $(-2, 4)$, $(1, -3)$, $(-1.7, 1)$ for particle 1, particle 2 and particle 3. The particles are assumed to be circular and produce a scattering pattern the same as that shown in Figure 11a. The scattering paths for the three particles are shown in Figure 12a. The deviation of scattering angle from the specular path is shown in Figure 12b. The red area is the scattering distribution of the entire surface and the others are for the scattering distribution of the local particles. It shows that for a rotationally symmetrical surface, the local defects have a similar distribution to the overall scattering distribution. Therefore, with the directional statistics of the scattering that contributes the stray light, a BRDF measurement aiming at the scattering of a single particle can be arranged. By inserting the measured data, the scattering caused by defects and reaching the detector is $1.34 \times 10^{-10}$ W, $3.54 \times 10^{-10}$ W, $1.09 \times 10^{-9}$ W, respectively, for unit incident power. Compared with $1.05 \times 10^{-5}$ W caused by roughness scattering, the stray light produced

by such three particles is much smaller than the roughness scattering and can be ignored. However, with an accumulation of such particles, the image quality can be reduced.

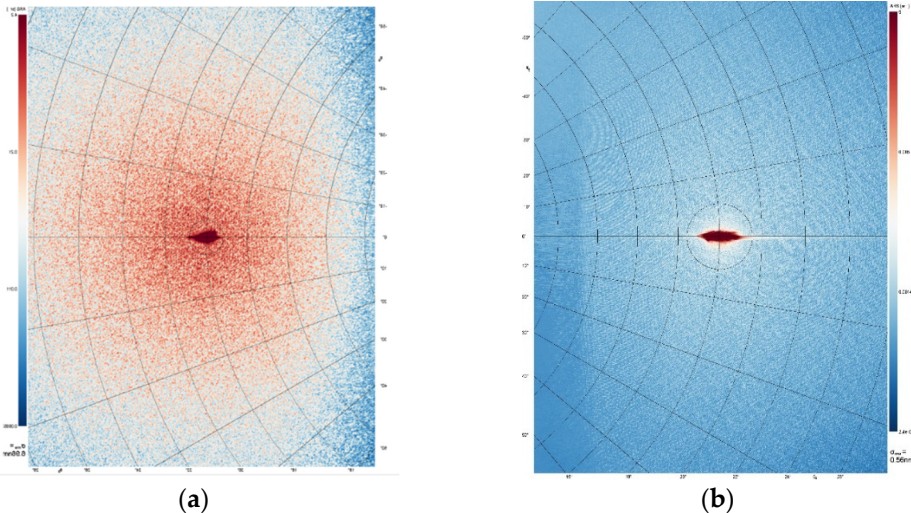

**Figure 11.** The scattering image of small particles on the superpolished sample (**a**) and the instrument signature (**b**).

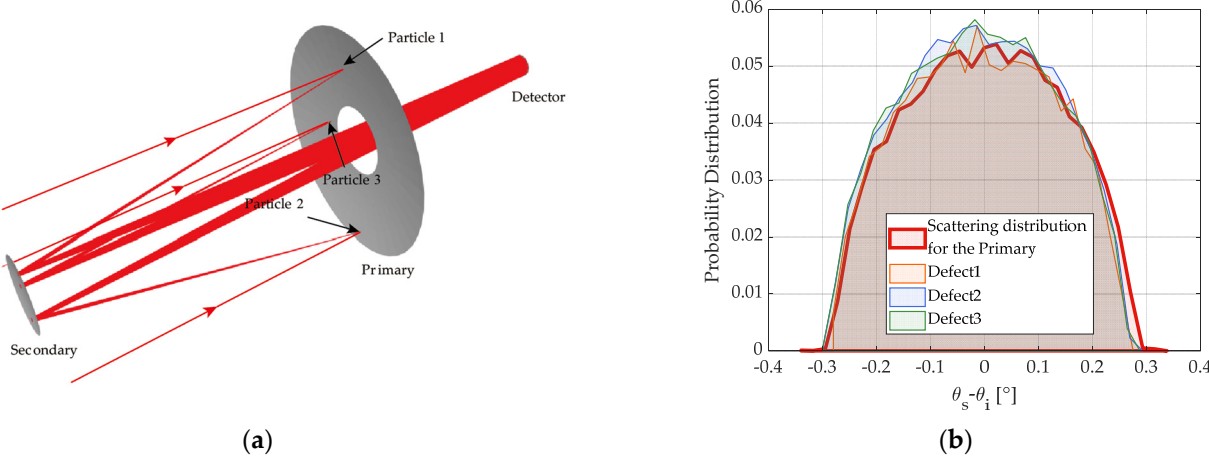

**Figure 12.** The scattering caused by local particles. (**a**) The optical path of the scattering which filled the entire detector. (**b**) The probability distribution of angular deviation from the specular direction due to small particle scattering.

## 4. Discussion

The imperfections of an optical surface lead to scattering and are an important source of stray light. However, the selection of the layout of an optical system is also critical and should be considered simultaneously. Due to the stochastic nature of surface imperfections and the diversity of optical systems, this paper discusses a method to determine the effective optical range of the scattering. The directional information of the delivered scattering is recorded. Based on the experimental data, the directional information on the one hand can determine the effective spatial frequency range, and the band-limited rms roughness can hence determine the effect of typical surface structures such as intrinsic texture, waviness, etc. The results show that the effective frequency range is different for different optical elements in a system even through the same surface is assumed. On the other hand, the effect of local defects such as a single particle or scratch is also discussed. By means of the ray tracing method discussed in this paper, the averaged angle of incidence and the range of scattered angle can be determined, which provides valuable information for scattering measurement. For example, for the mentioned system, the scattering measurement can be

conducted at the angle of incidence of 5° and the scattered angle ranges from −20° to 20°. In this way, it is no longer necessary to conduct many anisotropic scattering measurements.

In the future, more typical surface structure as a result of figuring, polishing and coating will be considered. The accumulation of contamination will also be considered. Furthermore, the polarized scattering can be measured and polarization ray tracing can be applied to analyze the polarization reaction of the optical system.

**Author Contributions:** Conceptualization, R.L.; Data curation, S.W.; Formal analysis, R.L.; Funding acquisition, Z.W.; Investigation, Z.C.; Methodology, Z.C.; Project administration, Z.W.; Resources, S.W. and C.F.; Software, R.L.; Supervision, C.F.; Writing—original draft, R.L.; Writing—review and editing, R.L. All authors have read and agreed to the published version of the manuscript.

**Funding:** This research was funded by National Natural Science Foundation of China, grant number 62075214; National Key R&D Program of China, grant number 2020YFC2200104.

**Data Availability Statement:** The data in publicly available for asking.

**Conflicts of Interest:** The authors declare no conflict of interest.

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
