# Peer review of "Effective Optical Scattering Range Determination Based on Ray Tracing"

_applsci, doi:10.3390/app13010307_

Round 1
Reviewer 1 Report
Very interesting paper, definitely something I would personally be interested to use in the future. The paper is technically sound and well organized, however english could be improved in terms both of grammar and spelling. Here are my main remarks (some other english mistakes might remain):
Line 15 : saying that the « optical system decides” is not rigorous, replace by something like “the optical system layout determines”
Line 39: “for a peridic produced” , missing a word, for example a periodic structure?
Line 39: “scattering that can pass” should be “that is able to reach the detector”
Line 40: “small angle”, would be better “scattering near the specular direction”
Line 41: do you mean like that does not reach the detector? If yes then indeed it results in energy loss, but that does not mean a contrast reduction
Line 43-44: rephrase sentence, hard to understand
Line 74: “this is no unique”, replace this by “there is no unique”
Line 76: incorrect English sentence
Line 137: “scattered rays will be selected” to be rephrased
Figure 5:we see drops in the profile for each curve, I assume this is an experimental artefact, is this the case or do you have any reason to believe it is a physical effect? If it is an artefact, why don’t you remove the point from the data?
Line 215: first two sentences should be written with more care
Line 217: “scattering of the typical Cassegrain” to be replaced by something like stray light from scattering occurring in a typical Cassegrain …
Line 240 to 244: hard to understand, rephrase
Figure 9: is it evaluated angularly as a function of the position on the mirror? Is this the distribution for all the rays hitting the detector, neglecting rays which don’t end up to the detector? This is what I understand but it is not obvious from the text. Be more clear about that in the text
Figure 10: on what data is the fitted curve actually fitted? Is it on the blue dots representing the measurement? If yes how can you find the periodic peak
Line 271: “determination the spatial …”, incorrect phrase, should be “determining the spatial …“
Line 278: such path could be verified experimentally and precisely identified with the time of flight method which allows to determine stray light origin by temporally discriminating the light at the detector during stray light testing. Add comment and cite the following corresponding paper https://www.nature.com/articles/s41598-021-89324-y
Line 316: what do you mean by selection of the optical system? I assume you mean that you want to improve the imperfections of the optical surface based on the layout of the optical system. Rephrase sentence so that it is clearer
Reviewer 2 Report
In this paper the authors deal with the important problem of surface irregularities on polished optical disks. They use a new method of ray tracing to simulate light scattering and apply their method to a Cassegrain telescope configuration. The work is valuable and certainly deserves to be published. There are, however, some points to be clarified and improved.
- It is not emphasized that which data are based on actual measurement and which one are a result of simulation. I am especially confused in the case of dust particles.
- In Fig. 5, there are dips in the BRDF curves at the angles, which corresponds to the direction of the incoming beam. Is this an artifact?
- What is the advantage of the proposed method in comparison to the WLI method?
- In Figs 7 and 8 it is not possible to follow the path of the rays, because there are to many of them. Maybe it would be better to pick out one or two rays and indicate which one is connected to primary or secondary scattering.
- The composition of several sentences are unclear, furthermore the numbering of figures is sometimes wrong. The authors should check the manuscript before publishing it.
Author Response
- It is not emphasized that which data are based on actual measurement and which one are a result of simulation. I am especially confused in the case of dust particles.
All of the topographic and scattering properties are based on real measured data, the influence is on the system is based on ray tracing.
- In Fig. 5, there are dips in the BRDF curves at the angles, which corresponds to the direction of the incoming beam. Is this an artifact?
The drops are simply due to the block of incident beam. It is quite normal for BRDF measurement. It is automatically removed during the implementation to the optical system. To remove the ambiguity, I remove the drops.
- What is the advantage of the proposed method in comparison to the WLI method?
The scattering measurement directly reflect the properties of optics. For example, the working wavelength is 532nm of a certain optical system. For WLI, the result is always from white light. However, we can directly use 532nm for scattering measurement which is certainly more accurate.
- In Figs 7 and 8 it is not possible to follow the path of the rays, because there are to many of them. Maybe it would be better to pick out one or two rays and indicate which one is connected to primary or secondary scattering.
Thanks for your advice, the manuscript has been changed according to your suggestions.
- The composition of several sentences are unclear, furthermore the numbering of figures is sometimes wrong. The authors should check the manuscript before publishing it.
Thanks for your advice, the manuscript has been changed according to your suggestions.